# Phylogenetic Analyses of *Hydnobolites* and New Species from China

**DOI:** 10.3390/jof8121302

**Published:** 2022-12-15

**Authors:** Shan-Ping Wan, Lan-Lan Huang, Meng-Jin Cui, Cheng-Jin Yu, Wei Liu, Rui Wang, Xiao-Fei Shi, Fu-Qiang Yu

**Affiliations:** 1College of Resources and Environment, Yunnan Agricultural University, Kunming 650201, China; 2The Germplasm Bank of Wild Species, Yunnan Key Laboratory for Fungal Diversity and Green Development, Kunming Institute of Botany, Chinese Academy of Sciences, 132 Lanhei Road, Kunming 650201, China

**Keywords:** Ascomycota, hypogeous fungi, *Pezizaceae*, phylogeny, taxonomy

## Abstract

*Hydnobolites* is an ectomycorrhizal fungal genus with hypogeous ascomata in the family *Pezizaceae* (*Pezizales*). Molecular analyses of *Hydnobolites* using both single (ITS) and concatenated gene datasets (ITS-nLSU) showed a total of 223 sequences, including 92 newly gained sequences from Chinese specimens. Phylogenetic results based on these two datasets revealed seven distinct phylogenetic clades. Among them, the ITS phylogenetic tree confirmed the presence of at least 42 phylogenetic species in *Hydnobolites*. Combined the morphological observations with molecular analyses, five new species of *Hydnobolites translucidus* sp. nov., *H. subrufus* sp. nov., *H. lini* sp. nov., *H. sichuanensis* sp. nov. and *H. tenuiperidius* sp. nov., and one new record species of *H. cerebriformis* Tul., were illustrated from Southwest China. Macro- and micro-morphological analyses of ascomata revealed a few, but diagnostic differences between the *H. cerebriformis* complex, while the similarities of the ITS sequences ranged from 94.4 to 97.2% resulting in well-supported clades.

## 1. Introduction

The genus *Hydnobolites* Tul. and C. Tul., with *H. cerebriformis* Tul. as the type species, was first described in 1843 [1]. It was characterized as having generally lobed or folded ascomata with a pseudoparenchymatous cortex that could be changed to a loose hyphal area toward the hymenium. The hymenium was poorly arranged around veins that led to the surface with openings generally between folds of the ascomata. Ellipsoid to pyriform, eight-spored asci were irregularly arranged between canals in the medullary area and whereas globose spores with strongly projecting spines were scattered within the asci [1,2,3,4,5,6].

*Hydnobolites* has a complex, intertwined taxonomic history. The genus was previously placed in the *Tuberaceae* [7,8,9,10,11,12], and then in the *Terfeziaceae* [3,5,13,14,15] or in the *Pyronemataceae* [16]. Trappe [5] kept some hypogeous lines, as families, within his *Pezizales*, while other hypogeous taxa were placed alongside epigeous species in various mixed families, he hence regarded *Hydnobolites* to be close to both *Pachyphloeus* and *Terfezia*. Such classifications were later tested in a longlasting study based on both cytological and ultrastructural features of asci and ascospores by Kimbrough et al. in 1991 [17] and further clarified in 1994 [18], led to the placement of *Hydnobolites* in the *Pezizaceae*. The systematic delimitation of this genus belonging to *Pezizaceae* has become clear with the molecular evidence in recent phylogenetic studies [19,20,21,22].

There are three *Hydnobolites* species accepted in the Dictionary of Fungi [23], and 18 species are listed in the current Index Fungorum online database, while five out of them are now reclassified to other families and genera, such as *Discinaceae* and *Tuberaceae* (http://www.indexfungorum.org/Names/Names.asp (accessed on 13 December 2022)). To date, the commonly accepted species include a European *H. cerebriformis*, an American *H. californicus* and four species from China: *Hydnobolites baodingensis* E. Wu and Z. Lan [24], *H. canaliculatus* L. Fan, Meng Chen and Ting Li, *H. roseus* L. Fan, Meng Chen and Ting Li, *H. shanxiensis* L. Fan, Meng Chen and Ting Li, and *H. yunnanensis* L. Fan, Meng Chen and Ting Li [25]. Among them, *H. baodingensis* is considered to be a species of *Mattirolomyces* rather than *Hydnobolites* according to the morphological description and ecological habits [25,26]. Xu reported the distribution of *H. cerebriformis* in Tibet, China in 2003 [27], however, which could not be effectively identified based on molecular evidence. Recently, the sequences of ectomycorrhiza (ECM) from Gansu and voucher specimen from Hubei, showed that *H. cerebriformis* was indeed distributed in China [25].

Geographically, *Hydnobolites* species distribute across Europe, North America, and Asia and form mycorrhizal symbioses with both broad leaf and conifer trees to benefit the growth and survival of host plants [21,25,26,28]. By including sequences derived from environmental samples in phylogenetic analyses, spore mats and ECM root tips, these data contributed to geographic distribution and habitat profiles for the genus *Hydnobolites* and also revealed a greater diversity than was previously known [19,21,25,29,30,31,32,33,34,35,36].

Recently, several specimens of *Hydnobolites* were collected in southwest China. We thus compared these specimens with previously recorded species of *Hydnobolites* and showed them to represent different new species, which led to the description of four new species and a new record of *H. cerebriformis* for China [2,3,24,25,27]. In addition, geographical distribution and species diversity of this genus are also clarified and reassessed.

## 2. Materials and Methods

### 2.1. Sampling and Morphological Observations

Specimens were collected from Yunnan province, Sichuan province and Tibet Autonomous Region in Southwest China. The information of each species is described in Appendix A. Macroscopic descriptions were based on detailed field notes made on fresh ascomata. Microscopic characters were later examined from fresh and dried materials following the methods of Yang and Zhang [37]. All new species were described in detail, including ascomata, peridium, gleba, paraphyses, asci and ascocarps. For the new record species *H. cerebriformis*, ascospores characteristics were not included due to its low maturity.

Hand-cut sections were mounted in 5% (*w*/*v*) KOH and examined under a light microscope (LM, Leica DM2500, Leica Microsystems, Wetzlar, Germany), while potassium hydroxide (2–5%) Congo red was employed to enhance contrast in microscopical observations. For evaluation of the range of spore size, 60 ascospores were measured from the specimen. Measurements of ascospores are given as (a)b–c(d), where b–c includes a minimum of 90% of the measured values. Extreme values (a and d) are given in parentheses. The abbreviation ‘Q’ represents the range of ratio of spore length to spore width calculated for each spore and “Q_m_ ± av” for the average Q of all spores ± sample standard deviation. For scanning electron microscopy (SEM), spores were scraped from the dried gleba onto double-sided tape (NISSHIN-EM, Japan), and this was mounted directly on an SEM stub, coated with gold-palladium, and examined and photographed using a JSM-5600LV SEM (JEOL, Tokyo, Japan). The specimens are deposited at the Herbarium of Cryptogams, Kunming Institute of Botany, Chinese Academy of Sciences (HKAS) or Yunnan Agricultural University (YNAU).

### 2.2. Molecular Identification

Total DNA was extracted from pieces of dried ascomata with a modified CTAB procedure [38]. Universal primer pairs ITS1F/ITS4 [38,39] and LROR/LR5 [40] were used for the amplifications of the internal transcribed spacers 1 and 2 with the 5.8S rDNA (ITS) and the large subunit of the nuclear ribosomal region (nrLSU), respectively. New sequences are listed in Appendix A highlighted in bold characters.

Polymerase chain reactions (PCR) were performed using the following procedure: 25 µL of PCR reaction solution contained 1 μL DNA, 1 μL (5 μm) of each primer pair, 2.5 μL 10 × buffer (Mg^2+^plus), 1μL dntps (1 mM), 0.5 μL BSA (0.1%), 0.5 μL MgCl_2_, 1 U of Taq DNA polymerase (Takara Tag, Takara Biotechnology, Dalian, China). PCR reactions were run as follows: 94 °C for 5 min, followed by 35 cycles of 94 °C for 30 s, 52 °C for 1 min and 72 °C for 1 min. The final reaction was followed by an extension at 72 °C for 10 min. The PCR products were sent to Tsingke Biotechnology Co., Ltd. (Beijing, China) for purifying and sequencing.

### 2.3. Phylogenetic Analyses

An ITS and an ITS-nrLSU combined dataset were used to illustrate and identify the phylogenetic position of new species in this genus. Representatives of the ITS and nrLSU sequences of *Hydnobolites* species available in GenBank were retrieved and combined with our dataset. An ITS alignment of *Hydnobolites* was built including all lineages identified in the most recent phylogenetic works, as well as all sequences available originated from environmental samples (Appendix A). Sequences derived from *Delastria* were selected and employed as outgroups. ITS and ITS-nrLSU alignments were made using the online version of the multiple sequence alignment program MAFFT v7 [41], applying the L–INS–I strategy, and were manually adjusted in BioEdit v.7.1.3.0 [42]. Alignments of all datasets used in this study were submitted to TreeBASE (No. 29971).

The phylogenetic relationships of taxa were inferred using maximum likelihood (ML) and Bayesian inference (BI). ML analyses on the ITS and ITS-LSU dataset were, respectively, performed in RAxML v.7.2.6 [43] and the GTR + GAMMA substitution model with parameters unlinked. ML bootstrap (BS) replicates (1000) were computed in RAxML with a rapid bootstrap analysis and search for the best-scoring ML tree. Bayesian analysis on the ITS and ITS-LSU datasets were performed in MrBayes v.3.1.2 [44] and the GTR + I + G model were selected as the best model under the Akaike Information Criterion (AIC) [45] implemented by MrModeltest v.2.3 [46]. Bayesian analysis was carried out using the selected model with four chains sampled every 100 generations and the average standard deviations of split frequencies were less than 0.01 at the end of the run and ESS (effective sampling size) values were > 200. A majority rule consensus tree was built after discarding trees from a 25% burning. Posterior probabilities (PPs) were calculated using the sumt command implemented in MrBayes. Nodes were annotated with support values from Maximum Likelihood and Bayesian analyses if at least one of these was considered significant. Support values were considered significant when ML bootstrap (BP) values were above 70% or posterior probability (PP) values were above 0.90. Trees generated by the two analyses were viewed and exported in FigTree v1.3.1.

## 3. Results

### 3.1. Molecular Phylogenetic Analyses

ML and BI analyses based on two datasets yielded identical tree topologies, respectively, the ITS tree inferred from the ML analysis and the ITS-nrLSU tree inferred from the BI analysis are shown (Figure 1 and Figure 2).

The ITS data matrix consisted of 147 taxa (including 46 sequences provided in this study) and 618 characters: 228 of ITS1 (complete), 148 of 5.8S (complete) and 242 of ITS2 (complete). ML of the general ITS dataset including all new sequences and those recovered from public databases (Figure 1) produced a phylogenetic tree consisting of seven distinct clades (Clade 1–Clade 7), six of them displayed significantly supported except for relatively lower supports of ML and strong supports of BI for Clade 1 (BP 65, PP 1.0).

Phylogenetic tree based on ITS dataset confirmed the presence of at least 42 phylogenetic species in genus *Hydnobolites* (Figure 1, red dots). Five new species clusters (*Hydnobolites translucidus*, *H. subrufus*, *H. lini*, *H. sichuanensis*, *H. tenuiperidius*) and one new record species (*H. cerebriformis*) were scattered in Clade 1 and Clade 3 together with species from Europe and North America. Moreover, ITS rDNA sequences analyses showed that species between *H. cerebriformis* complex (red thick branch, Clade 1) shared 94.4%–97.2% similarities, while the similarities between sequences of *H. cerebriformis* were 97.2%–100% (subclade 1). The clustering characteristics of species in the topologies were coincided with the similarities and differences of ITS. Another new species *H. sichuanensis* was separately formed an independent branch in Clade 1 with strong support (BP 100, PP 1.0). Clade 2 includes five North American species represented by *H*. *californicus* with effective support (BP 92 PP 1.0). As for Clade 3, *H. tenuiperidius* was sister to *H. roseus*, and DNA sequence analyses showed that *H. tenuiperidius* and *H. roseus* (MK192825, holotype) shared < 91.9% ITS similarities. Furthermore, there were another three phylogenetic species in the Clade 3, which were marked as subclades 2–4, ITS rDNA sequences analyses showed these species share < 96.1% similarities. Genetic information and topologies all indicated that subclades 2–4 were not the same species, this is contrary to the inference of previous studies [25]. Among the remaining Clades, Clades 4–5 are composed of American species while Clades 6–7 are composed of Chinese species.

ITS-nrLSU combined data matrix contained 79 taxa (including 46 sequences provided in this study) and 1419 characters after alignment and trimmed (618 of ITS and 801 of LSU). BI analysis of the ITS-nrLSU combined dataset including all new sequences and the available sequences from public databases yielded similar topologies with ITS analysis (Figure 2) except that clade 3 had no significant supports.

### 3.2. Taxonomy

#### 3.2.1. *Hydnobolites Translucidus* S.P. Wan and F.Q. Yu, sp. nov.

MycoBank: 834662.

Etymology: In terms of the translucent gleba of the fruiting bodies (Figure 3).

Holotype: China, Yunnan Province, 25°53′19″ N and 103°29′24″ E, in humic soil under a pure *Quercus guyavifolia* Levl. forest, at about 2433 m, 14 August 2016, wsp761 (HKAS95861, GenBank accession: ITS = MT174247, LSU = OP642324).

Diagnosis: The new species *Hydnobolites translucidus* has deeply infolded and fused, white, dirty-white ascomata; translucent and watersoaked gleba; spores comparatively larger (up to 25.5 μm in length and 25.0 μm in width) than in its close relative *H. cerebriformis* (19–20–22 μm) [3], smaller than *H. subrufus* (up to 26.5 μm in length and 25.3 μm in width) and *H. lini* (up to 26.9 μm in length and 26.1 μm in width). *Hydnobolites translucidus* has whitish ascomata while *H. subrufus* has a red hue surface. The thinnest value of the peridium of *H. translucidus* is larger than *H. subrufus* and *H. lini*.

Description: Ascomata irregularly globose, often grooved, cracked, deeply infolded and fused, soft but firm, surface smooth, glabrous to tomentose, minutely downy to scurfy in the depression, 0.5–2.0 cm in diam., white, dirty-white, translucent and locally light brown when fresh, golden brown to brown when dried. Odour none. Peridium 43.0–182.0 μm thick, hyphae hyaline, pseudoparenchymatous, composed of subglobose, ellipsoid, and polygonal cells of 3.2–37.0× 3.0–20.0 μm. Gleba solid, transparent, watersoaked white and stick when fresh, with whitish veins, convoluted and chambered, composed of intricately interwoven, hyaline and thin-walled hyphae, 4.0–13.0 μm diam, the cells irregularly subglobose, cylindrical interwoven to inflated, 3.0–16.0 × 3.0–12.5 μm. Paraphyses colorless, straight to bent, free from one another, septate, slightly swollen and blunt at tip, sparsely projecting to a distance of 25.0 μm above surface of peridium, 5.0–15.0 μm diam. Asci globose to subglobose, pyriform, ellipsoid or irregular, (74.0)75.0–106.0(–112.0) × (60.5–)63.0–92.8(–103.0) μm, hyaline, sessile or with a short stalk, 8 spored, crozier clearly seen in situ in immature asci. Ascospores spherical, white to light yellow, excluding their alveolate-reticulate ornamentation, (14.5–)16.0– 23.0(–25.5) × (14.0–)15.5–23.0(–25.0) μm, Q = 1.0–1.08 (Q_m_ ± 1.02), the ornaments of spores formed an alveolate-reticulum of 0.7–6.7 μm high, 3–7 meshes across the diameter when mature.

Habitat and distribution: Solitary to scattered in small groups, on humus-rich soil with dead leaves and wood under trees of *Quercus guyavifolia* and *Pinus armandii* Franch., where the fruiting bodies contacts with the soil. There are clumps of soil attached, only known from southwest China.

Additional material examined: China, Yunnan Province, 25°53′48″ N and 103°29′14″ E, in humic soil under a pure *Pinus armandii* forest, at about 2462 m, 14 August 2016, wsp758 (HKAS95859, GenBank accession: ITS = OM758120, LSU = OP642323). China, Yunnan Province, 25°58′19″ N and 102°43′32″ E, in soil under a *Pinus armandii* forest, at about 1920 m, 24 September 2021, wsp1435 (YNAU0568, GenBank accession: ITS = OM758130, LSU = OP642335). China, Yunnan Province, 25°61′15″ N and 100°25′28″ E, in soil under a *Pinus armandii* forest, 27 January 2022, wsp1919 (YNAU0988, GenBank accession: ITS = OP740750, LSU = OP799844). China, Yunnan Province, 25°61′37″ N and 100°25′54″ E, in soil under a *Pinus armandii* forest, 27 January 2022, wsp1920 (YNAU0989, GenBank accession: ITS = OP740751, LSU = OP799845). China, Yunnan Province, 25°61′55″ N and 100°25′73″ E, in soil under a *Pinus armandii* forest, 27 January 2022, wsp1923 (YNAU0990, GenBank accession: ITS = OP740752, LSU = OP799846).

Notes: *Hydnobolites translucidus* is characterized macroscopically by the characteristics of deeply infolded, fused, white to dirty-white ascomata, translucent and watersoaked gleba and microscopically by the morphology of its peridium and ascospores.

Morphologically, the new species *H. translucidus* is similar to its phylogeneticly close relatives *H. cerebriformis* and our newly described *H. subrufus* and *H. lini*. All species shared the same grooved, cracked, infolded, fused, soft but firm ascomata, smooth and glabrous surface, and downy depression. However, *H. cerebriformis* has smaller ascospores (19–20–22 μm), as well as asci (85–100 × 60–75 μm) [3], can be clearly distinguished from the other three species. Despite *H. translucidus* was very similar with *H. subrufus* and *H. lini* in the structure of pseudoparenchymatous peridium, the size and shape of asci and ascospores, but *H. translucidus* has whitish ascomata while *H. subrufus* has red to brownish red surface. Furthermore, the thinnest value of the peridium of *H. subrufus* and *H. lini* is smaller than that of *H. translucidus*, so the asci or spores of *H. subrufus* and *H. lini* can be seen directly from the dried ascomata surface. Phylogeneticlly, these four species all belong to the *H. cerebriformis* complex according to this study based on both ITS and ITS-nrLSU, respectively, and the distinctions of species were also supported by the phylogenetic analyses (Figure 1 and Figure 2).

#### 3.2.2. *Hydnobolites subrufus* S.P. Wan and F.Q. Yu, sp. nov.

MycoBank: 834663.

Etymology: In terms of the fruiting bodies surface with red tone (Figure 4).

Holotype: China, Yunnan Province, 25°53′50″ N and 103°28′35″ E, in humic soil under a pure *Quercus guyavifolia* forest, at about 2354 m, 14 August 2016, wsp769 (HKAS95869, GenBank accession: ITS = MT174248, LSU = OP642325).

Diagnosis: The diagnostic characteristics of the new species *Hydnobolites subrufus* are white, red to brownish red ascomata with thin peridium (20.0–244.0 μm). The convex shape of the asci can be directly observed from the surface of the dried ascomata, which can be well distinguished from *H. cerebriformis* and *H. translucidus. Hydnobolites lini* is well distinguished from *H. subrufus* by the clearly visible spores on the fruiting bodies surface.

Description: Ascomata irregularly ellipsoid to globose, grooved, cracked, infolded and fused, soft but firm, surface smooth, glabrous to tomentose, minutely downy to scurfy in the depression, 0.5–1.0 cm in diam., white to red when fresh, golden yellow to golden brown when dried and the surface showed obvious convex shape of the asci. Odor none. Peridium 20.0–244.0 μm thick, hyphae hyaline, pseudoparenchymatous, composed of subglobose, ellipsoid, and polygonal cells of 2.5–25.0 × 2.0–17.5 μm. Gleba solid, watery white when fresh, with whitish veins, grooved, cracked, composed of intricately interwoven, hyaline and thin-walled hyphae, 1.5–9.5 μm diam, the cells irregularly subglobose, square, cylindrical interwoven to inflated, 5.0–17.0 × 4.5–16.5 μm. Paraphyses colorless, straight to bent, free from one another, septate, slightly swollen and blunt at tip, sparsely projecting to a distance of 24.0 μm above surface of peridium, 5.0–10.0 μm diam. Asci globose to subglobose, pyriform, ellipsoid or irregular, (75.0–)80.0–105.0(–110.1) × (61.0–)67.0–93.5(–105.0) μm, hyaline, sessile or with a short stalk, 8 spored, crozier clearly seen in situ in immature asci. Ascospores spherical, white to light yellow, excluding their alveolate-reticulate ornamentation, (17.5–)18.5–25.5(–26.5) × (17.0–)18.0–25.3 μm, Q = 1.0–1.09 (Q_m_ ± 1.03), the ornaments of spores formed an alveolate-reticulum of 0.8–7.0 μm high, 3–7 meshes across the diameter when mature.

Habitat and distribution: Solitary to scattered in small groups, in humus-rich soil with dead leaves and wood under trees of *Quercus guyavifolia*, where the fruiting bodies contacts with the soil. There are clumps of soil attached, only known from southwest China.

Additional material examined: China, Yunnan Province, 25°53′59″ N and 103°28′26″ E, in humic soil under a pure *Quercus guyavifolia* forest, at about 2380 m, 14 August 2016, wsp769-1 (YNAU0971, GenBank accession: ITS = OM758121, LSU = OP642326).

Notes: Morphologically, the new species *Hydnobolites subrufus* is similar to *H. cerebriformis*, *H. lini* and *H. translucidus.* However, as described in the diagnosis and notes of *H. translucidus*, *H. subrufus* can be separated from *H. cerebriformis* [3] and *H. translucidus* in the color of ascomata, thickness of peridium and size of asci and spores*. Hydnobolites lini* differs from *H. subrufus* in its visible ascospores on the surface of ascomata. Another species (*H. roseus*) [25] described from northwest China also has red ascocarps, but differs from *H. subrufus* by its thicker peridium (55–325 μm), smaller asci (55–100 × 50–62.5 μm), larger spores extremum (up to 30 μm) and distant phylogenetic distance (Figure 1 and Figure 2).

#### 3.2.3. *Hydnobolites lini* S.P. Wan and F.Q. Yu, sp. nov.

MycoBank: 846353.

Etymology: In appreciation of Mr. Yong-Xiang Lin, who has contributed many samples to the research of genus *Hydnobolites* (Figure 5).

Holotype: China, Yunnan Province, 25°58′20″ N and 102°45′21″ E, in humic soil under a pure *Pinus armandii* forest, at about 2270 m, 29 October 2021, wsp1723 (YNAU0860, GenBank accession: ITS = OM758149, LSU = OP642357).

Diagnosis: The new species *Hydnobolites lini* has obvious asci and ascospores on the surface of dried ascomata when compared with the known species.

Description: Ascomata irregularly globose to subglobose, grooved, cracked, infolded and fused, soft but firm, surface smooth, glabrous to tomentose, downy densely in grooves and depression, 0.8–2.5 cm in diam., white when fresh, slightly reddish after a while, golden yellow to golden brown when dried. Odor none. Peridium 18.0–393.0 μm thick, hyphae hyaline, pseudoparenchymatous, composed of subglobose, columnar, and polygonal cells of 4.5–40.0 × 3.5–25.0 μm. Gleba solid, white when fresh, locally light brown, with whitish veins, grooved, cracked, composed of intricately interwoven, hyaline and thin-walled hyphae, 3.4–28.0 μm diam, the cells subglobose, square to cylindrical, swollen and interwoven, 4.0–32.3 × 3.2–25.4 μm. Paraphyses colorless, straight to bent, free from one another, septate, blunt at tip and always swollen, sparsely projecting to a distance of 40.0 μm above surface of peridium, 3.5–9.0 μm diam. Asci globose to subglobose, pyriform, ellipsoid or irregular, (75.0–)80.0–105.0(–110.1) × (61.0–)67.0–93.5(–105.0) μm, hyaline, sessile or with a short stalk, 8 spored, crozier clearly seen in situ in immature asci. Ascospores spherical, white to light yellow, excluding their alveolate-reticulate ornamentation, (15.2–)16.3–26.6(–26.9) × (14.8–)15.5–26.0(–26.1) μm, Q = 1.0–1.17 (Q± = 1.05), the ornaments of spores formed an alveolate-reticulum of 0.7–5.5 μm high, 3–7 meshes across the diameter when mature.

Habitat and distribution: Solitary, in humic soil with *Pinus armandii*, where the fruiting bodies contacts with the soil. There are clumps of soil attached, only known from southwest China.

Additional material examined: China, Yunnan Province, 26°67′98″ N and 103°39′22″ E, in humic soil under pure *Pinus armandii*, at about 2245 m, 22 August 2021, wsp1246 (YNAU0362, GenBank accession: ITS = OM758125, LSU = OP642330). China, Yunnan Province, 26°67′55″ N and 103°39′18″ E, in humic soil under pure *Pinus armandii*, at about 2159 m, 30 August 2021, wsp1257 (YNAU0373, GenBank accession: ITS = OM758126, LSU = OP642331). China, Yunnan Province, 26°46′32″ N and 103°26′91″ E, in humic soil under pure *Pinus armandii*, at about 2016 m, 22 September 2021,wsp1369 (YNAU0494, GenBank accession: ITS = OM758128, LSU = OP642333). China, Yunnan Province, 25°57′19″ N and 102°42′61″ E, in humic soil under trees dominated by *Pinus armandii* and associated with *Quercus* sp., at about 1899 m, 24 September 2021, wsp1433 (YNAU0566, GenBank accession: ITS = OM758129, LSU = OP642334). China, Yunnan Province, 25°57′08″ N and 102°42′34″ E, in humic soil under trees dominated by *Pinus armandii* and associated with *Quercus* sp., at about 1890 m, 24 September 2021, wsp1436 (YNAU0569, GenBank accession: ITS = OM758131, LSU = OP642336). China, Yunnan Province, 26°41′36″ N and 103°32′97″ E, in humic soil under *Pinus armandii*, at about 2073 m, 18 October 2021, wsp1635 (YNAU007, GenBank accession: ITS = OK598967, LSU = OP642343). China, Yunnan Province, 26°41′46″ N and 103°32′06″ E, in humic soil under *Pinus armandii*, at about 2011 m, 18 October 2021, 1641 (YNAU010, GenBank accession: ITS = OK598969, LSU = OP642344). China, Yunnan Province, 26°41′76″ N and 103°32′18″ E, in humic soil under *Pinus armandii*, at about 2022 m, 18 October 2021, wsp1642 (YNAU011, GenBank accession: ITS = OK598970, LSU = OP642347). China, Yunnan Province, 26°41′75″ N and 103°32′21″ E, in humic soil under *Pinus armandii*, at about 2019 m, 18 October 2021, wsp1642-2 (YNAU0774, GenBank accession: ITS = OM758141, LSU = OP642349). China, Yunnan Province, 26°41′06″ N and 103°32′79″ E, in humic soil under *Pinus armandii*, at about 1993 m, 18 October 2021, wsp1643-2 (YNAU0776, GenBank accession: ITS = OM758143, LSU = OP642351). China, Yunnan Province, 26°41′88″ N and 103°32′79″ E, in humic soil under *Pinus armandii*, at about 2004 m, 18 October 2021, wsp1643-4 (YNAU0778, GenBank accession: ITS = OM758144, LSU = OP642352). China, Yunnan Province, 26°16′91″ N and 102°49′60″ E, in humic soil under *Pinus armandii*, at about 1860 m, 29 October 2021, wsp1690 (YNAU0825, GenBank accession: ITS = OM758145, LSU = OP642353). China, Yunnan Province, 26°16′88″ N and 102°49′58″ E, in humic soil under *Pinus armandii*, at about 1862 m, 29 October 2021, wsp1691 (YNAU0826, GenBank accession: ITS = OM758146, LSU = OP642354). China, Yunnan Province, 26°16′86″ N and 102°49′55″ E, in humic soil under *Pinus armandii*, at about 1866 m, 29 October 2021, wsp1693 (YNAU0828, GenBank accession: ITS = OM758147, LSU = OP642355). China, Yunnan Province, 26°16′87″ N and 102°49′51″ E, in humic soil under *Pinus armandii*, at about 1868 m, 29 October 2021, wsp1694 (YNAU0829, GenBank accession: ITS = OM758148, LSU = OP642356).

Notes: The new species *Hydnobolites lini* differs from its close relative *H. cerebriformis, H. subrufus* and *H. translucidus* by the obvious asci and ascospores on the surface of dried ascomata. The other newly described species *H. tenuiperidius* in this paper also has the same characteristic but differs from *H. lini* by its smaller ascospores ((13.3–22.0(–23.0) × (12.5–)–22.0(–22.6) μm)) and distant phylogenetic distance (Figure 1 and Figure 2).

#### 3.2.4. *Hydnobolites sichuanensis* S.P. Wan and F.Q. Yu, sp. nov.

MycoBank: 846354

Etymology: In reference to the location of the type collection (Figure 6).

Holotype: China, Sichuan Province, 30°05′22″ N and 100°78′53″ E, in humic soil under *Abies chensiensis* Tiegh. and *Picea wilsonii* Mast. forests, at about 3911 m, 10 October 2021, wsp1575 (YNAU 0705, GenBank accession: ITS = OM758132, LSU = OP642337).

Diagnosis: Differs from other species by possessing smaller spores ((15.0–)16.0–21.0 × (14.5–)15.5–20.5 μm) and thicker peridium (80.0–390.0 μm).

Description: Ascomata irregularly globose to subglobose, grooved, cracked, lobed and infolded, soft but firm, surface smooth but tomentose locally, downy densely to scurfy in the depression, 0.7–0.9 cm in diam., white to light red and light orange when fresh, golden yellow to golden brown when dried. Odor none. Peridium 80.0–390.0 μm thick, hyphae hyaline, pseudoparenchymatous, composed of subglobose, ellipsoid, and polygonal cells of 5.5–41.0 × 4.0–34.0 μm. Gleba solid, watery white when fresh, with whitish veins, grooved, cracked, composed of intricately interwoven, hyaline and thin-walled hyphae, 1.4–9.8 μm diam, the cells irregularly subglobose, square to cylindrical, swollen and interwoven, 3.4–43.5 × 3.4–34.0 μm. Paraphyses colorless, subglobose to cylindrical, straight to bent or knobby, sometimes bulbous, free from one another, septate, blunt at tip and always swollen, sparsely projecting to a distance of 36.5 μm above surface of peridium, 3.8–10.5 μm diam. Asci globose to subglobose, pyriform, ellipsoid or irregular, (55.6–)62.0–83.5(–85.5) × (47.5–)52.5–74.5(–77.0) μm, hyaline, sessile or with a short stalk, 8 spored, crozier clearly seen in situ in immature asci. Ascospores spherical, white to light yellow, excluding their alveolate-reticulate ornamentation, (15.0–)16.0–21.0 × (14.5–)15.5–20.5 μm, Q = 1.0–1.08 (Q_m_ ± 1.02), the ornaments of spores formed an alveolate-reticulum of 0.7–6.7 μm high, 3–5 meshes across the diameter.

Habitat and distribution: Solitary, in humus-rich soil with dead leaves of *Abies chensiensis* and *Picea wilsonii*, where the fruiting bodies contacts with the soil. There are clumps of soil attached, only known from southwest China.

Additional material examined: China, Sichuan Province, 30°05′45″ N and 100°78′89″ E, in humic soil under trees dominated by *Quercus guyavifolia* and associated with *Picea wilsonii*, at about 3932 m, 10 October 2021, wsp1582 (YNAU0712, GenBank accession: ITS = OM758133, LSU = OP642338).

Notes: *Hydnobolites sichuanensis* is characterized microscopically by the morphology of its peridium and ascospores. The peridium are thicker (80.0–390.0 μm) than the known species. The ascospores of *H. sichuanensis* are regular spherical and relatively smaller ((15.0–)16.0–21.0 × (14.5–)15.5–20.5 μm) when compared with all known species except *H. californicus*, which has smaller ascospores (14–18 μm) and bigger alveoli (3–4 meshes) [2]. Furthermore, in phylogenetic analysis, *H. sichuanensis* forms a independently distinct branch with significant support (BP 100, PP 1.0) (Figure 1 and Figure 2).

#### 3.2.5. *Hydnobolites tenuiperidius* S.P. Wan and F.Q. Yu, sp. nov.

MycoBank: 846355

Etymology: In terms of the thin-peridiumed Peridium (Figure 7).

Holotype: China, Yunnan Province, 25°07′18″ N and 102°50′24″ E, in humic soil under a pure *Pinus armandii* forest, at about 2222 m, 14 November 2021, wsp1751 (YNAU0899, GenBank accession: ITS = OM758150, LSU = OP642358).

Diagnosis: Differs from other species by possessing extremely thin peridium (5.5–222.8 μm).

Description: Ascomata irregularly globose to subglobose, grooved, cracked, lobed and infolded, soft but firm, surface smooth, glabrous to tomentose, minutely downy in grooves and depression, 2.1–2.8 cm in diam., white when fresh, light red after a while, golden yellow to golden brown when dried. Odour none. Peridium 5.5–222.8 μm thick, hyphae hyaline, pseudoparenchymatous, composed of subglobose, columnar, and polygonal cells of 4.5–40.0 × 3.5–25.0 μm. Gleba solid, white when fresh, with whitish veins, grooved, cracked, locally with white continuous interwoven mycelia, colorful after dry, gleba composed of intricately interwoven, hyaline and thin-walled hyphae, 4.5–24.4 μm diam, the cells subglobose, square to cylindrical, 4.7–44.6 × 3.7–30.0 μm. Paraphyses colorless, straight to bent, sometimes bulbous, free from one another, septate, blunt at tip and always swollen, sparsely projecting to a distance of 50.5 μm above surface of peridium, 5.5–18.6 μm diam. Asci globose to subglobose, pyriform, ellipsoid or irregular, (50.6–)55.0–98.3(–99.0) × (37.8–)40.0–89.0(–90.0) μm, hyaline, sessile or with a short stalk, 8 spored, crozier clearly seen in situ in immature asci. Ascospores spherical, white to light yellow, excluding their alveolate-reticulate ornamentation, 13.3–22.0(–23.0) × (12.5–)–22.0(–22.6) μm, Q = 1.0–1.18 (Q± = 1.03), the ornaments of spores formed an alveolate-reticulum of 0.6–7.3 μm high, 3–7 meshes across the diameter.

Habitat and distribution: Solitary, in humic soil with *Pinus armandii*, where the fruiting bodies contacts with the soil. There are clumps of soil attached, only known from southwest China.

Additional material examined: China, Yunnan Province, 25°07′32″ N and 102°50′25″ E, in humic soil under pure *Pinus armandii*, at about 2238 m, 14 November 2021, wsp1752 (YNAU0900, GenBank accession: ITS = OM758151, LSU = OP642359). China, Yunnan Province, 25°07′15″ N and 102°50′19″ E, in humic soil under pure *Pinus armandii*, at about 2256 m, 14 November 2021, wsp1753 (YNAU0901, GenBank accession: ITS = OM758152, LSU = OP642360). China, Yunnan Province, 25°09′66″ N and 100°63′92″ E, in humic soil under pure *Pinus armandii*, at about 2256 m, 14 November 2021, wsp1829 (YNAU0986, GenBank accession: ITS = OP740748, LSU = OP799842). China, Yunnan Province, 25°09′61″ N and 100°63′ 90″ E, in humic soil under pure *Pinus armandii*, at about 2250 m, 14 November 2021, wsp1830 (YNAU0987, GenBank accession: ITS = OP740749, LSU = OP799843).

Notes: *Hydnobolites tenuiperidius* is characterized microscopically by the morphology of its peridium and ascospores. Some peridium areas are extremely thin (5.5 μm), so the spores can be clearly observed from the surface of the dried specimen. This characteristic was also shared by *H. lini*, but *H. lini* has larger size of asci ((75.0–)80.0–105.0(–110.1) × (61.0–)67.0–93.5(–105.0) μm) and spores ((15.2–)16.3–26.6(–26.9) × (14.8–)15.5–26.0(–26.1) μm).

In our phylogenetic analyses, six specimens of *H. tenuiperidius* form a single branch with strong molecular support (BS = 100, PP = 1.0) and clusters into the subclade 2 that includes *H. roseus* (Figure 1). The ITS similarities between *H. tenuiperidius* and *H. roseus* (ITS = MK192825, Holotype) are less than 91.9%. Morphologically, despite *H. tenuiperidius* shares the rose color ascomata and similar ascospores size characters with *H. roseus*, but *H. tenuiperidius* can be separated from *H. roseus* by the characteristics of downy depression, thin peridium and 3–7 meshes, while *H. roseus* has nearly glabrous depression, thicker peridium, and 3–5 meshes across the spores [25]. In addition, it should also be noted that there also two specimens (BJTC FAN770 and HMAS81912) were identified as *H. roseus* [25], but they were clustered into a single branch and distinctly separated from the holotype of *H. roseus,* and they share less than 95.1% ITS similarities with the holotype sequence of *H. roseus,* so the molecular evidence suggests they might not be the same species.

#### 3.2.6. *Hydnobolites cerebriformis* Tul., Tul., Annales des Sciences Naturelles Botanique 19: 378 (1843) [1] (Figure 8)

MycoBank: 2389

Ascomata irregularly globose, grooved, cracked, much lobed and infolded, 0.7 to 1.0 cm diam., waxy, transparent partially, surface smooth, glabrous to tomentose, downy in the depression, at first white, then yellowish to sandy colored when fresh, pale to light brown when dried. Odor none. Peridium 53.8–317.0 μm thick, hyphae hyaline, pseudoparenchymatous, composed of subglobose, ellipsoid, and polygonal cells of 1.7–34.6 × 1.7–19.7 μm. Gleba solid, white when fresh, with indistinct few whitish veins, connecting with folds, hyaline and thin-walled hyphae, 0.9–16.6 μm diam, the cells cylindrical interwoven to inflated, 2.0–24.8 × 1.9–20.7 μm. Paraphyses colorless, straight to bent, free from one another, septate, slightly swollen or blunt at tip, sparsely projecting to a distance of 65.0 μm above surface of peridium, 1.5–12.7 μm diam. Asci globose to subglobose, pyriform, ellipsoid or irregular, 83.0–96.5(–101.8) × 62.1–72.0(–75.5) μm, hyaline, sessile or with a short stalk, crozier clearly seen in situ in immature asci, immature spores are occasionally visible.

Habitat and distribution: Solitary, in humic soil with *Pinus wallichiana* A. B. Jackson and associated with *Rosa* sp., where the fruiting bodies contacts with the soil, there are clumps of soil attached. The specimen was collected in August, but it is still immature. Know from Tibet of China. Furthermore, according to the ITS sequences of ECMs and specimens in the public database, this species is widely distributed in Asia and Europe, including China (Hubei and Gansu provinces), Japan, Estonia, UK and Latvia.

Material examined: China, Tibet, 27°46′36″ N and 88°90′48″ E, in humic soil under trees of *Pinus wallichiana* and associated with *Rosa* sp., at about 3187 m, 4 August 2021, wsp1171 (YNAU0318, GenBank accession: ITS = OM758123, LSU = OP642328). China, Tibet, 27°46′37″ N and 88°90′51″ E, in humic soil under trees of *Pinus wallichiana* and associated with *Rosa* sp., at about 3185 m, 4 August 2021, wsp1171-1 (YNAU0972, GenBank accession: ITS = OM758124, LSU = OP642329).

Notes: *Hydnobolites cerebriformis* is originally described from Europe, and also reported from North America [3,12,17,18,21]. The occurrence of *H. cerebriformis* in China is confirmed from Tibet under *Pinus wallichiana* dominant forest based on molecular and morphological evidence. Despite this, the spores’ maturity was low and has not been described in detail. The other characteristics of *H. cerebriformis* observed in this study conformed to those described by previous workers [2,8,12,17,20]. On this basis, we further illustrated and described the characteristics of paraphyses.

A specimen (HMAS60271) ITS sequence (MK236552) from Hubei Province and an ITS sequence (LC623535) from the ECM of *Pinus tabulaeformis* Carr. in Gansu Province of China also match this species, in accordance with previous results [25]. Furthermore, ITS sequence (AB218135) from the ECM of *Abies homolepis* Siebold and Zucc. also confirmed the distribution of this species in Japan of Asia. In general, sequences from different regions were clustered together as *H. cerebriformis* with significant supports (subclade 1, BP 100, PP 1.0), and the ITS similarities between these sequences were up to 97.2–100%, well demonstrated that they are conspecific and widely distributed.

## 4. Discussion

*Hydnobolites*, a relatively small and infrequent genus [28], is still poorly understood across the world, although its study can be traced back to 1834 [1]. The high species diversity and broadly geographic distribution of specimens and ECMs suggest that *Hydnobolites* has wider hosts and ecological adaptability [19,21,31,32,33,34,35,36,37,38,39,40,41,42,43,44,45,46,47,48,49,50,51,52,53,54,55,56,57,58,59,60,61,62,63,64].

Our phylogenetic analysis based on an ITS and ITS-nrLSU databases revealed that *Hydnobolites* species clustered into seven distinct clades (Figure 1 and Figure 2). Among them, Clade 1 received sub-significant supports in the ML analysis but had significant supports in the Bayesian results (Figure 1 BP 65, PP 1.0; Figure 2 BP 57, PP 0.99). The other clades were well supported by effective supports in the phylogenetic trees, except clade 3 deriving from the ITS-nrLSU analysis (Figure 1 and Figure 2). Specifically, the phylogenetic position and topologies of Clades 1 and 3 were similar with previous results [25]. Furthermore, Clade 1 had greater Bayesian supports (Figure 1 PP 1.0, Figure 2 PP 0.99) when compared with the previous results [25], which could be due to the increase of sampling in present study. Clade 3 inferred from the ITS analysis was supported by strong statistical bootstrap from ML and posterior probability from BI (Figure 1 BP 93, PP 1.0), this was also consistent with the result of Li et al. [25]. However, Clade 3 inferred from ITS-nrLSU dataset was not supported by significant values, which probably due to the insufficient phylogenetic signal of the nrLSU analyzed in present work, or gaps in the nrLSU phylogenetic diversity because of an incomplete sampling.

Phylogenetic trees based on ITS dataset also confirmed the presence of at least 42 phylogenetic species distributed in Asia, Europe and North America (Figure 1). Of these, ten are identified as known species, including one new record for China that was also shared across Japan, Europe and North America, and five species were identified as new species from this study. Other phylogenetic species were not named since they were from ECMs or a lack of ascus material.

Although there were subtle morphological differences, it might not be easy to define the close relationship between *H. cerebriformis* complex. The molecular evidence showed that significant divergences were observed in the ITS region. The ITS-nrLSU combined dataset was used to further determine their relationships within the complex, where different species of *H. cerebriformis* complex were clearly and consistently identified.

Previous research showed that *Hydnobolites* morphological characters were brain-like and had pale ascomata, thin excipulum, pseudoparenchymatous peridium, a lack of paraphyses or were without well-differentiated paraphyses, an apparent lack of basal mycelial tuft, globose spores with a complete reticulum ornamentation, 8 spores per mature ascus, and a tendency to turn orange after dry [12,20,65]. In consistency with these above-mentioned characteristics, we also observed well-differentiated paraphyses, clearly visible spores and ascus on the surface and basal mycelial tuft adhering to soil. The morphological characteristics of this genus have been hence further understood by this present study.

         Key to the taxa of *Hydnobolites*
1.Ascomata whitish to yellowish or light brown.................................................................................................................................................21.Ascomata whitish to reddish...............................................................................................................................................................................32.Ascomata downy...................................................................................................................................................................................................42.Ascomata glabrous................................................................................................................................................................................................53.Asci or ascospores can be observed from the surface of ascomata................................................................................................................63.Asci or ascospores can not be observed from the surface of ascomata.........................................................................................................74.Ascospores have larger sizes (14.5–)16.0–23.0(–25.5) × (14.0–)15.5–23.0(–25.0)..................................................................*H. translucidus*4.Ascospores have medium sizes (19–20–22 μm)..................................................................................................................*H. cerebriformis* [3]4.Ascosporeshave smaller sizes (14–18 μm).............................................................................................................................*H. californicus* [2]5.Ascospores have larger sizes (17.5–32.5 μm)....................................................................................................................*H. canaliculatus* [25]5.Ascospores have medium sizes (18.75–26.25 μm)..............................................................................................................*H. shanxiensis* [25]5.Ascospores have smaller sizes (17.5–25 μm).....................................................................................................................*H. yunnanensis* [25]6.Ascospores have smaller sizes (13.3–22.0(–23.0) × (12.5–)–22.0(–22.6) μm).......................................................................*H. tenuiperidius*6.Ascospores have larger sizes...............................................................................................................................................................................87.Ascomata downy, ascospores smaller (up to 21.0 × 20.5 μm)................................................................................................*H. sichuanensis*7.Ascomata glabrous, ascospores larger (up to 30 μm)..................................................................................................................*H. roseus* [25]8.Ascomata have obvious red color.....................................................................................................................................................*H. subrufus*8.Ascomata white to slightly reddish...........................................................................................................................................................*H. lini*

## Figures and Tables

**Figure 1 jof-08-01302-f001:**
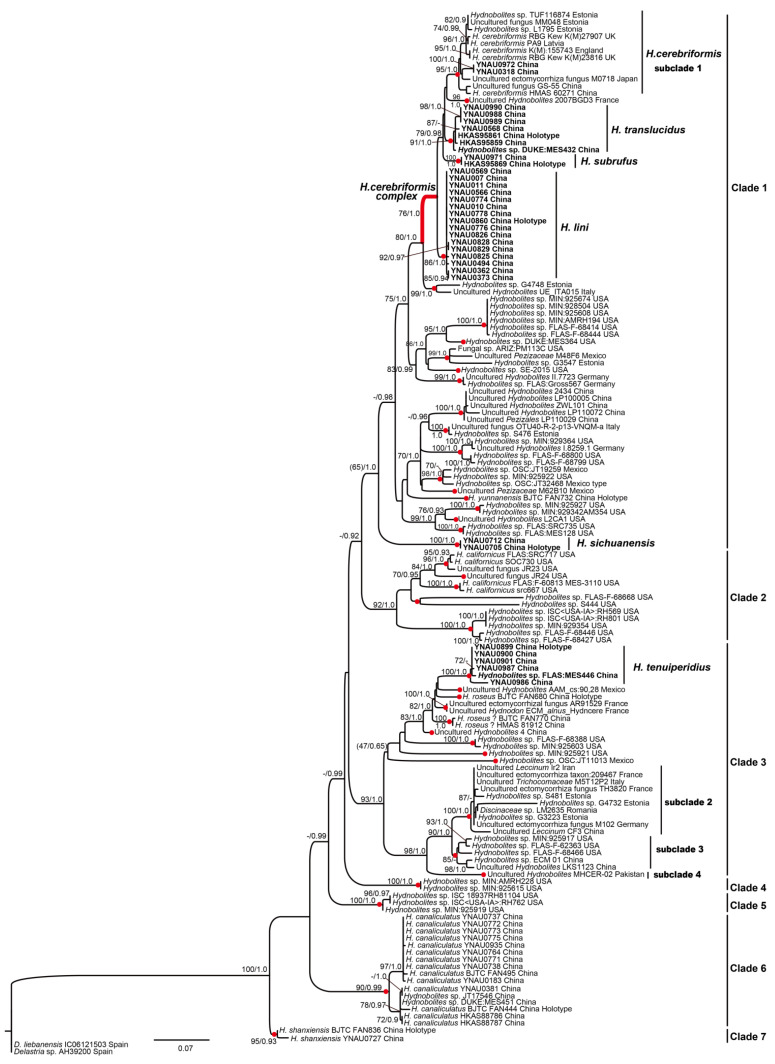
The consensus phylogram of the Genus *Hydnobolites* obtained in RAxML of ITS rDNA. Nodes were annotated if supported by > 70% ML BP or > 0.90 Bayesian PP, but non-significant support values are exceptionally represented inside parentheses. New species are in bold font. Complex is reported as red thick branch.

**Figure 2 jof-08-01302-f002:**
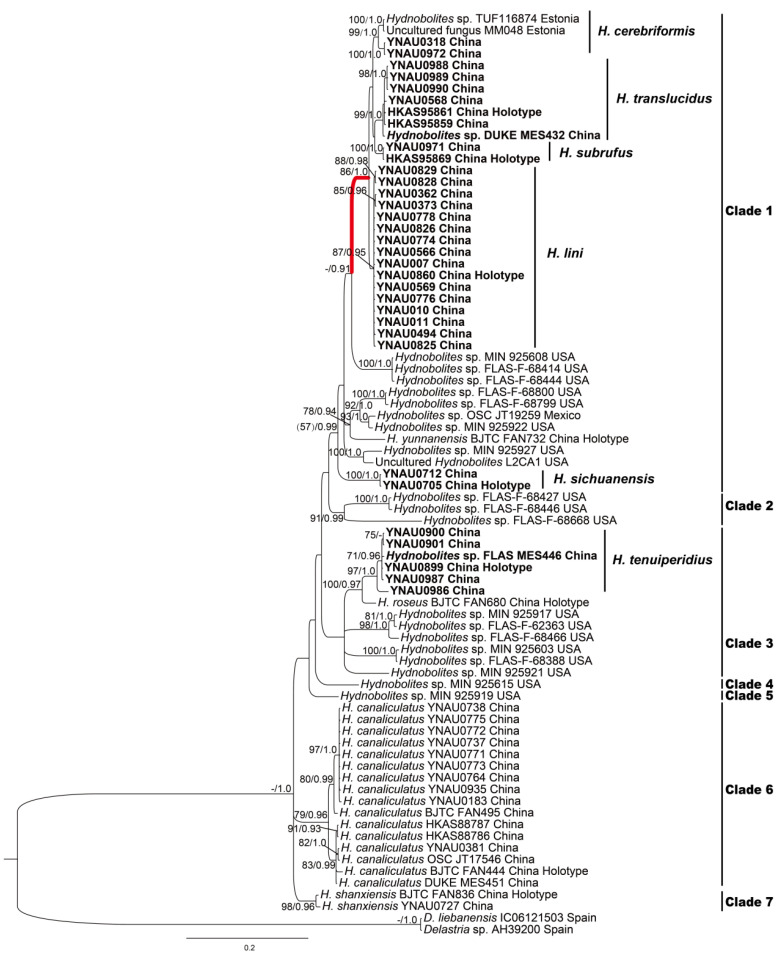
The consensus phylogram of the Genus *Hydnobolites* obtained in MrBayes of ITS-nrLSU. Nodes were annotated if supported by > 70% ML BP or > 0.90 Bayesian PP, but non-significant support values are exceptionally represented inside parentheses. New species are in bold font. Complex is reported as red thick branch.

**Figure 3 jof-08-01302-f003:**
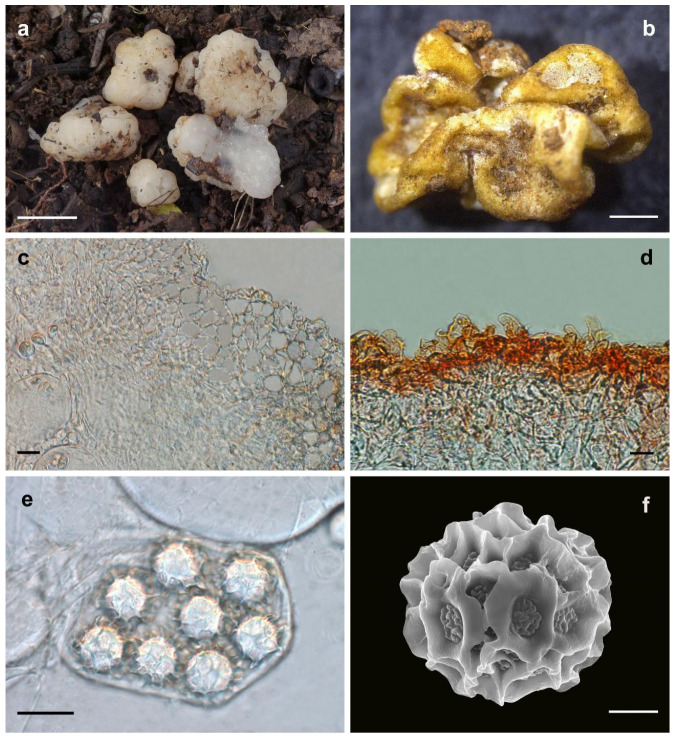
*Hydnobolites translucidus* (Holotype, HKAS95861). (**a**) Fresh ascomata and gleba; (**b**) dried ascoma; (**c**) peridium; (**d**) paraphyses; (**e**) asci and ascospores under LM; (**f**) ascospore under SEM. Scale bars: a = 1 cm; b = 1 mm; c–e = 20 μm; f = 5 μm.

**Figure 4 jof-08-01302-f004:**
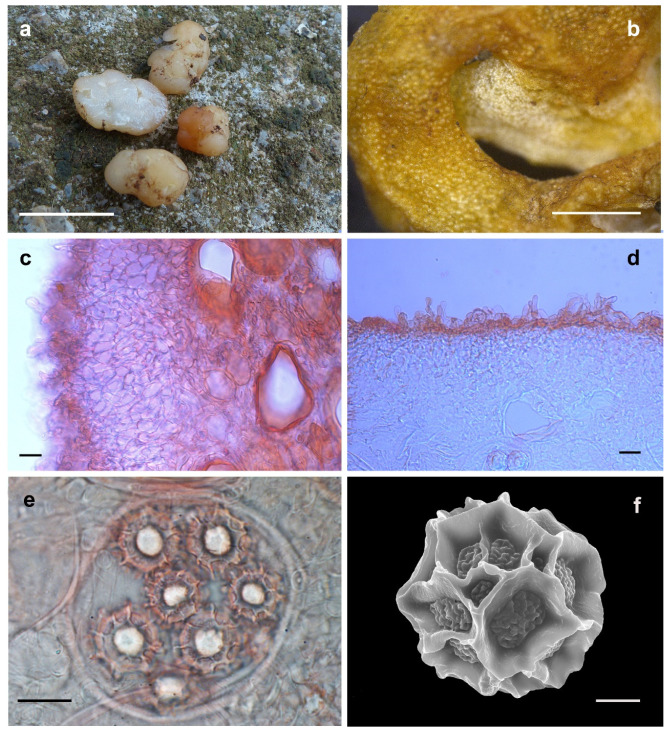
*Hydnobolites subrufus* (Holotype, HKAS95869). (**a**) Fresh ascomata and gleba; (**b**) dried ascoma; (**c**,**d**) peridium and paraphyses; (**e**) asci and ascospores under LM; (**f**) ascospore under SEM. Scale bars: a = 1 cm; b = 1 mm; c–e = 20 μm; f = 5 μm.

**Figure 5 jof-08-01302-f005:**
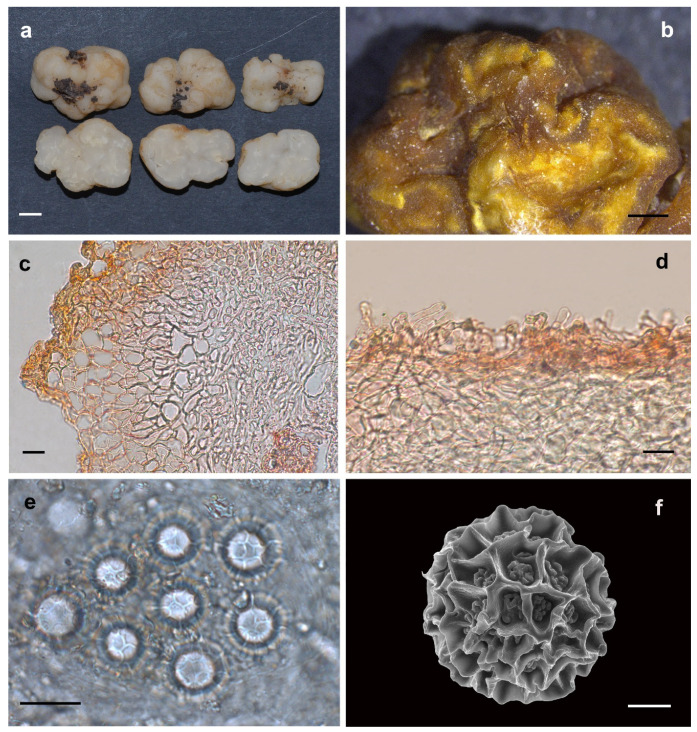
*Hydnobolites lini* (Holotype, YNAU0860). (**a**) Fresh ascomata and gleba; (**b**) dried ascoma; (**c**) peridium; (**d**) paraphyses; (**e**) ascus and ascospores under LM; (**f**) ascospore under SEM. Scale bars: a = 1 cm; b = 1 mm; c–e = 20 μm; f = 5 μm.

**Figure 6 jof-08-01302-f006:**
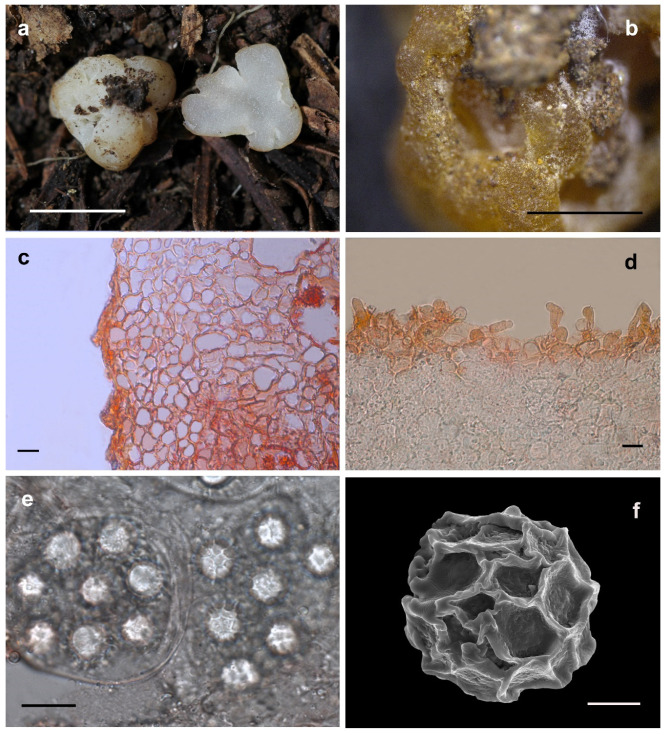
*Hydnobolites sichuanensis* (Holotype, YNAU0705). (**a**) Fresh ascoma and gleba; (**b**) dried ascoma; (**c**) peridium; (**d**) paraphyses; (**e**) asci and ascospores under LM; (**f**) ascospore under SEM. Scale bars: a = 1 cm; b = 1 mm; c–e = 20 μm; f = 5 μm.

**Figure 7 jof-08-01302-f007:**
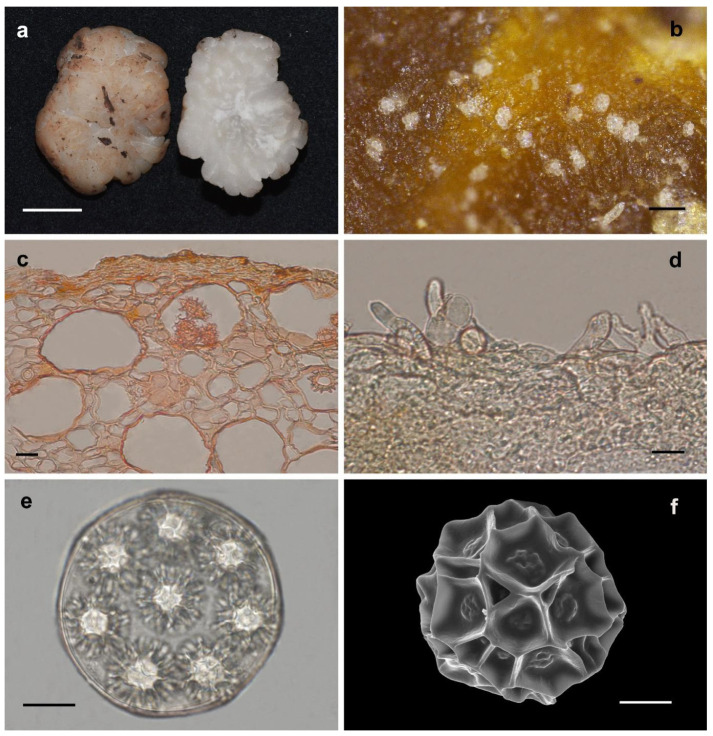
*Hydnobolites tenuiperidius* (Holotype, YNAU0899). (**a**) Fresh ascoma and gleba; (**b**) surface of dried ascoma; (**c**) peridium; (**d**) paraphyses; (**e**) ascus and ascospores under LM; (**f**) ascospore under SEM. Scale bars: a = 1 cm; b = 0.1 mm; c–e = 20 μm; f = 5 μm.

**Figure 8 jof-08-01302-f008:**
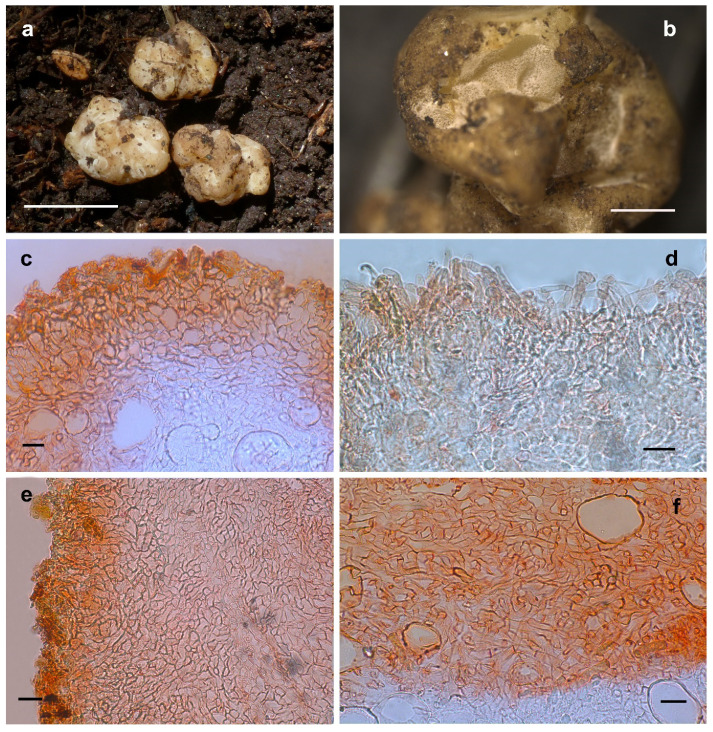
*Hydnobolites cerebriformis* (YNAU0318). (**a**) Fresh ascomata; (**b**) dried ascoma; (**c**,**e**) peridium; (**d**) paraphyses; (**f**) hyphae of gleba. Scale bars: a = 1 cm; b = 1 mm; c–f = 20 μm.

## Data Availability

A publicly available dataset was analyzed in this study. The resulting alignments were deposited in TreeBASE (http://www.treebase.org; accession number 29971 (accessed on 9 December 2022)). All newly generated sequences were deposited in GenBank (https://www.ncbi. nlm.nih.gov/genbank/ (accessed on 08 November 2022), mentioned in the text, Table and in Figure 1 and Figure 2). All new taxa were deposited in MycoBank (https://www.mycobank.org/ (accessed on 13 December 2022)).

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
