# Peer review of "Phylogenetic Analyses of Hydnobolites and New Species from China"

_jof, 2022, doi:10.3390/jof8121302_

Round 1

Reviewer 1 Report

Review ref jof 2070999 Wan et al. Phylogenetic analyses of Hydnobolites and new species from China

The authors performed taxonomic analysis of five new species in the genus Hydnobolites.
In the introduction, the species concept is introduced, material and methods is complete, molecular description of new species reliable on basis of maybe to scarce molecular data. The statistical support of phylogenetic relationships statistcal support is sometimes low (e. g. clade 3). The molecular results may have been benefited when different regions of the data sets of ITS and LSU rDNA would have been tested for tree estimation. Considering the inclusion of secondary barcodes could have helped to increase clade support. The morphological description is formally complete containing a full set of compared characters, with almost always sound and complete diagnosis and description.
The inclusion of uncultured environmental sequences increases our knowledge about distribution but also diversity of taxa. This really increases the quality and informativeness of this manuscript. However, sometimes environmental sequences are rather short, either targeted on ITS1 or 2 gene regiones and prone to sequencing errors introduced by high throughput sequencing methods, in turn leading to overestimation of phylogenetic distance and/or leading to decreased support of clades.

Abstract, Inroduction and parts of Results  are difficult to read because of language issues, the authors should consider editing the language thoroughly. I tried to give some suggestions for improvement of paragraphs in the first sections of the manuscript but had to give up. It is just to labourious, because of to many language mistakes. Therefrore I concentrated on the scientific content for the rest of the mansucript.

specific comments

L 11 remove second "Correspondence:"
L 14 datasets
- remove available
L 15 evidence seems to be not the right word here. Re-structure the sentence Molecuar analyses of the two datasets revealed...
L 18/19 what is a record species? re-structure the sentence Six new species isolated (?) from Southwest China were described and one species complex H...was proposed
L 21 ....while the similarity ITS sequences ranged from X to X % resulting in well-supported clades
L 44 names
L 49 H. cerebriformis was not described from China. Better write a total of 5 species was described from China. However, as you are talking about the species described in the genus, you should not mix reported and described species here. The correct phrase should read:... and a total of 5 species was described from China:...list without H. cerebriformis.
L 54 "EcM" The mycorrhiza first mentioned here, please explain the acronym at first use
L 62-65 Make shorter sentences:
"Recently, several specimens of Hydnobolites were collected in southwest China. We  compared these specimens with the previously recorded species of Hydnobolites, which led to the description of four new species and a new record of H. cere. for China."

L 67-70 Please check the language!

L 78 what means fully described here? Please explain, as not all readers might be familiar with Pezizales taxonomy.

L 90 please indicate the brand of the tape
L 90
L 97 "pieces" how large? how many g?
L 104 dNTPs
L 104-108 please use correct abbreviations for the units! It is µl not ML for microliter!
L 115 published works? List the original publications  of retrieved sequences them e.g. in a supplemental table, if possible.
L 116 Table; why did you choose Delastria as outgroup? Please explain in short, how distant is it from Hydnobolites?
L 118 If you use online ressources, indicate the last accession date (when you performed your analyses)
L 120 I could neither find your study nor an alignment under this number in TreeBase!

L 140 two datasets
L 141 why did you show ML trees instead of the MrBayes trees?
L 148-154 This rather belongs to discussion
L 154 can you further specify this? Which sequences have had which indels or whatever differences in comparison within this clade? You could further discuss this in the discussion section. You could have excluded to variable gene regions of your ITS rDNA data. Furthermore, the inclusion of more genetic loci could have helped to resolve phylogeny in clade 3.

Fig.1 e.g. "Uncultured Hydnobolites 2007BGD3 France" The inclusion of uncultured environmental sequences increases our knowledge about distribution but also diversity of taxa. However, sometimes environmental sequences are rather short, either targeted on ITS1 or 2 gene regiones and prone to sequencing errors introduced by high throughput sequencing methods, in turn leading to overestimation of phylogenetic distance. Therefore, sequences of uncultured should be checked for quality. How did you do that and what were your qualitiy criteria to include an environmental sequence?

L 180 Your outgroup taxa seem to be very distantly related to your genus, studied. Did this have had impacts on the branch lengths of your tree?

L 198  ff. Diagnosis: You are describing spore sizes as "up to" From this I would understand that the size of spores overlaps in certain stages of spore ripening? Thus this might not be the key diagnostic character for accurate species identification in Hydnobolites? What would you assign as the key diagnostic character then? Ascomata morphology? Can you explain the character "visibility/obious ascopores on the surface" to some more extent?

L 329 "...diameter when..."

L 413 known

L 468 "six specimens of H. tenuiperidius form a single branch"

L 540 what about adding more genetic loci, e g. protein coding genes to you analyses? Experimentally, elucidating the loci with best phylogenetic signal? I can see, that your descriptions of new species are backed up with ITS/LSU sequences but the support of this could possibly be increased by adding more genetic loci. Your discussion would benefit from this critical outlook.

Reviewer 2 Report

I feel this paper is well written and the five Hydnobolites taxa from China are probably indeed new species.

In my opinion, to add an up to date identification key (dichotomous) to all known Hydnobolites species or Hydnobolites species known from China would increase the value of the manuscript. In addition, to give a comparative table for the main morphological characters of the species separated within the H. cerebriformis complex would be also useful.

I have found only some typos and grammar errors in the manuscript (please find my suggestions in the reviewers copy).

Author Response

Respected reviewer:

Based on your professional and kind comments, our manuscript "Phylogenetic analyses of Hydnobolites and new species from China" has been revised.

Here below is our description on revision. Thank you very much.

Response to Reviewer 2 Comments

Comments and Suggestions for Authors

I feel this paper is well written and the five Hydnobolites taxa from China are probably indeed new species.

Point 1: In my opinion, to add an up to date identification key (dichotomous) to all known Hydnobolites species or Hydnobolites species known from China would increase the value of the manuscript. In addition, to give a comparative table for the main morphological characters of the species separated within the H. cerebriformis complex would be also useful.

Point 2: I have found only some typos and grammar errors in the manuscript (please find my suggestions in the reviewers copy).

Response: Thank you very much for your professional and kind suggestions. A identification key was added in the revised manuscript. Besides, typos, grammar errors and italics for names of all taxa regardless of rank have been corrected.

In addition, the manuscript was revised thoroughly. Special attention was paid to correct the former mistakes English. Prof. Xinhua He, who used to work in the University of Western Australia, was invited to correct the inappropriate expressions and improve the writing. We hope with these efforts the paper is more neatly prepared, concise and eadable.

Thank you very much for all of the constructive suggestions, helpful comments, and kind corrections.

Sincerely yours,

Shan-Ping Wan and Fu-Qiang Yu